# Efficient Certification for Probabilistic Robustness

## Abstract

Recent developments on the robustness of neural networks have primarily emphasized the notion of worst-case adversarial robustness in both verification and robust training. However, often looser constraints are needed and some margin of error is allowed. We instead consider the task of probabilistic robustness, which assumes the input follows a known probabilistic distribution and seeks to bound the probability of a given network failing against the input. We focus on developing an efficient robustness verification algorithm by extending a bound-propagation-based approach. Our proposed algorithm improves upon the robustness certificate of this algorithm by up to $8\times$ while with no additional computational cost. In addition, we perform a case study on incorporating the probabilistic robustness verification during training for the first time.

## 1 Introduction

Neural networks have found great success in a wide variety of applications. For many of these applications, understanding when and how a neural network can fail is crucial. Szegedy et al. (2014) found that almost visually imperceptible perturbations of input images could drastically change the output of a neural network. This realization set off a large area of research, both in finding ways of attacking neural networks and developing defense and certification methods against said attacks. The most common setting which is considered is the *worst-case* robustness given an $l_p$ norm. There have been a number of adversarial robustness certification methods which operate by finding provable $l_p$ balls for which the output of the neural network is guaranteed to be constant. In other words, they find lower bounds on the minimum $l_p$ radius for which an adversarial attack exists. This is done by relaxing the network for a bounded domain. Convex relaxations are typically used (Salman et al., 2019), although some works also consider quadratic program formulations (Raghunathan et al., 2018). Note that these are input-specific, so they do not give guarantees about the entire input domain.

As opposed to the worst-case adversarial robustness, there is also great interest in the threat model where the primary concern is natural noise and corruption rather than a malicious adversary and thus the perturbations follow some probability distribution. Typical neural networks have been found to be vulnerable to common corruptions (Dodge & Karam, 2016; Geirhos et al., 2018). We distinguish between these two notions of robustness as *adversarial robustness* and *probabilistic robustness*. We note that the terms such as corruption robustness, probability of violation, and adversarial density have also been used to refer to the general concept of probabilistic robustness. For the threat model of natural noise and corruption, there has not been as much work developed as for worst-case adversarial robustness. One of the very first probabilistic verification algorithms without imposing assumptions on the decision boundary is known as the PROVEN algorithm (Weng et al., 2019). In PROVEN, the authors derive probabilistic verification bounds based on existing worst-case robustness verification such as Fast-Lin and CROWN (Weng et al., 2018; Zhang et al., 2018).

**Contributions.** In this work, we generalize the PROVEN algorithm proposed by Weng et al. (2019) to infinite support and greatly improve the tightness of the robustness certificate *without* additional computation cost. We name our algorithm **I-PROVEN**, as the proposed

algorithm is a significantly improved version of PROVEN algorithm. The **I-PROVEN** algorithm can achieve significant improvements ($2\times$-$8\times$ tighter) on the tightness of probabilistic robustness certificate for both MNIST and CIFAR-10 models without additional computation cost, and it also enables the certification of probability distributions with infinite support. Based on our proposed algorithm, we conduct a case study on augmenting an existing training pipeline with probabilistic robustness verification bounds, and we find mixed results for the training. We examine potential causes and implications.

## 2 BACKGROUND AND RELATED WORKS

**Notation.** In order to describe related work in depth, we will lay out some notation. We define a $K$-layer feed-forward ReLU neural network with $n_0$ inputs and $n_K$ outputs $f : \mathbb{R}^{n_0} \to \mathbb{R}^{n_K}$ as

$$f(x) = f^{(K)}(x)$$
$$f^{(i+1)}(x) = W^{(i+1)}\sigma(f^{(i)}(x)) + b^{(i+1)}$$
$$f^{(1)}(x) = W^{(1)}x + b^{(1)}$$

In other words, $f^{(i)}(x)$ denotes the vector of pre-activation values in the $i^{\text{th}}$ layer. Generally, we work in the setting of image classifiers where a class $c$ is classified over a class $i$ if $f_c(x) - f_i(x) > 0$. To simplify notation, we assume that the neural networks $f$ which we are working with have already had this margin function applied to it for some given $c, i$. In other words, we assume $n_K = 1$ for convenience and we are interested in when $f(x) > 0$.

### 2.1 ADVERSARIAL ROBUSTNESS VERIFICATION

Adversarial robustness verification asks, given a neural network $f$ and a region $\mathcal{R}(\epsilon)$ in the input space, does there exist an $x \in \mathcal{R}(\epsilon)$ such that $f(x) \leq 0$? To solve this problem, we can formulate it as an equivalent optimization problem: $\min_{x \in \mathcal{R}(\epsilon)} f(x)$. If no such $x$ such that $f(x) \leq 0$ exists, or equivalently, if the minimum $f(x)$ is positive, then $f$ is robust for region $\mathcal{R}(\epsilon)$. If we can prove that $f$ is robust on regions $\mathcal{R}(\epsilon)$ for all $\epsilon \leq \underline{\epsilon}$, then the *robustness certificate* is $\underline{\epsilon}$. The robustness certificate is a lower bound for the true minimum distortion $\epsilon^*$.

The regions $\mathcal{R}(\epsilon)$ of general interest are $L_p$ balls $B_p(x_0, \epsilon)$ for a given image or input $x_0$. This arises from the interpretation that an adversary is perturbing $x_0$ by at most $\epsilon$ under a given $L_p$ norm. Note that certification only informs robustness about a single image. As far as we know, it is infeasible to certify an entire dataset other than processing it image by image.

**Convex relaxation for provable verification.** These methods find a convex relaxation of a neural network in order to find provable certifications for its adversarial robustness. We will discuss these methods in detail as our method builds on them in certain ways. There are a number of works following these methods (Weng et al., 2018; Singh et al., 2018; Zhang et al., 2018; Singh et al., 2019b), and a general framework for them is described in (Salman et al., 2019). We will use the setup used in CROWN (Zhang et al., 2018). In these methods, inequalities on pre-activation neurons are recursively computed

$$l^{(j)} \leq A_L^{(j)}x + b_L^{(j)} \leq f^{(j)}(x) \leq A_U^{(j)}x + b_U^{(j)} \leq u^{(j)}$$

for each layer $j$. Note that these are element-wise bounds, $l^{(j)}$ and $u^{(j)}$ are scalar vectors, and $A_L^{(j)}, b_L^{(j)}, A_U^{(j)}, b_U^{(j)}$ are linear transformations of inputs $x$. These linear bounds are obtained by relaxing the non-linear activation functions to linear lower and upper bounds given that the inputs to the activation functions are within some interval found from the inequalities applied to earlier layers, $l^{(i)}, u^{(i)}, i < j$. These inequalities are propagated backwards through the network until the original input is reached. Under the typical $l_p$ ball threat model, Hölder's inequality can give scalar bounds on these layers and the process can continue to the final outputs. (Singh et al., 2019a; Tjandraatmadja et al., 2020) have made progress on improving these bounds beyond the convex relaxation gap pointed out by Salman et al. (2019) by considering the activation functions on multiple neurons jointly.

## 2.2 Probabilistic robustness

For probabilistic robustness, we are considering a known probability distribution $D : \mathbb{R}^n \to [0, 1]$ which the inputs $x$ are sampled from. We will focus on additive iid uniform noise which we denote, in an abuse of notation, as $B_\infty(x, \epsilon)$. In other words, $B_\infty(x, \epsilon)$ is the distribution generated by sampling points evenly from the hyperrectangle $[x_1 - \epsilon, x_1 + \epsilon] \times [x_2 - \epsilon, x_2 + \epsilon] \times \cdots \times [x_n - \epsilon, x_n + \epsilon]$.

Then the problem of probabilistic robustness verification is to verify that

$$Pr_{x \sim D}[f(x) > 0] \geq 1 - Q \tag{1}$$

for some given failure probability $Q$.

We can define the robustness certificate similarly to how it was done for adversarial robustness. Weng et al. (2019) and Anderson & Sojoudi (2020) particularly consider the maximum $\epsilon$ parameterizing $D$ for which the above holds and we also provide such results. This is found by binary searching over $\epsilon$, as empirically we find that the robustness is monotonic in $\epsilon$ for the distributions we consider, but we note that there are no theoretical guarantees for this.

**Sampling methods.** Sampling gives well-established statistical guarantees which can be applied to the problem of probabilistic robustness. By using the neural network essentially as a black-box, Chernoff bounds can estimate the probability of the neural network giving the incorrect classification given a distribution. This has the advantage of making no assumptions on the model or the distribution, but requires a large number of samples to achieve a high degree of accuracy and there is an inherent uncertainty present in the application of such an algorithm. Baluta et al. (2021) notes for example, that proving that the probability is between $[0.1 - 0.5 \times 10^{-4}, 0.1 + 0.5 \times 10^{-4}]$ with a confidence of 0.9 would require $5.5 \times 10^6$ samples. To overcome this, they propose a framework which reduces the number of samples necessary, although this is dependent on the true probability. Anderson & Sojoudi (2020) also provide a method that can find upper bounds on the probability that a model is incorrect with a small number of samples. Webb et al. (2018) uses a clever sampling method that leverages the layered structure of common architectures. They require upwards of $10^7$ samples but are able to obtain precise estimations. Though they are unable to provide theoretical guarantees, they show that empirically, their estimations agree with naive Monte Carlo estimates with as many as $10^{10}$ samples.

## 2.3 Training adversarially-robust models

Training methods for improving adversarial robustness have generally taken two paths. The first augments the data with adversarial attacks in order to strengthen a model's resistance to such attacks (Madry et al., 2017). The second approach adds loss regularization terms that help the model learn robust features of the data. Xiao et al. (2018) identifies weight sparsity and ReLU stability as important factors in a model's adversarial robustness and builds a training framework which incorporates these. Other works use certification methods as regularization terms in order to improve the certifiable robustness of a model. Interval bound propagation (IBP) has found great success in this despite being a relatively loose certification method (Gowal et al., 2019). In particular, the efficiency of IBP has made it amenable to training. A number of other works have made progress in closing this efficiency gap (Zhang et al., 2019; Xu et al., 2020; Shi et al., 2021; Boopathy et al., 2021).

## 3 Our main results

In section 3.1, we illustrate the idea of deriving tighter probabilistic robustness certificate and provide the details of our **I-PROVEN** algorithm. We remark on alternative setups in section 3.2. In section 3.3, we conduct a case study on including our proposed probabilistic verification bounds into standard training. All experimental results are reported in section 4.

### 3.1 I-PROVEN: Improving the tightness of PROVEN algorithm

In this subsection, we will show how we could build on top of the state-of-the-art PROVEN algorithm (Weng et al., 2019) to derive tighter probabilistic robustness certificate in **I-PROVEN**. Using convex relaxation methods, one can find linear bounds with respect to the input such that,

$$A_L^{(K)}x + b_L^{(K)} \leq f(x) \leq A_U^{(K)}x + b_U^{(K)}, \quad \forall x \in \text{supp}(D). \tag{2}$$

PROVEN argues that by applying probabilistic bounds

$$\Pr_{x \sim D}[A_L^{(K)}x + b_L^{(K)} \geq 0] \geq 1 - q, \tag{3}$$

we can conclude that $f(x) \geq 0$ with probability at least $1 - q$. Notably, probabilistic inequalities are only considered in the final layer. Thus, they are able to give a probabilistic robustness certification.

To extend this method, we must look into how these linear bounds $A_L, b_L, A_U, b_U$ were obtained. This is done recursively: every layer's pre-activation neurons are bounded by linear bounds with respect to the input. Then, scalar bounds are obtained using Hölder's inequality on the linear bounds and the support of the input. These scalar bounds define intervals in the domain of each activation function for the next layer. This allows linear bounds to be calculated over the activation function and for the next layer to continue propagating linear bounds.

The key observation is that these linear bounds in previous layers can also apply with some probability rather than strictly and that the failure probabilities accumulate linearly. Assume that we have some $q'$ for which

$$\Pr_{x \sim D}[f(x) \geq A_L^{(K)}x + b_L^{(K)}] \geq 1 - q'. \tag{4}$$

Then a simple union bound gives that the probability of $f(x) \geq 0$ is at least $1 - q - q'$. This is a simple scenario which can be extended to involve all layers of the network.

**Theorem 1.** *In the convex-relaxation framework, for each scalar inequality $i$ ($l \leq A_L x + b_L$ or $u \geq A_U x + b_U$), denote $q_i$ as the probability that this inequality is violated with respect to the probability distribution $D$ which $x$ is sampled from. Then the probability of the final output of the convex-relaxation algorithm holding for a given $x \sim D$ is $\geq 1 - \sum_i q_i$.*

*Proof.* The convex-relaxation framework operates by making a series of $m$ inequalities which ultimately lead to the output layer. We can label these $L_1, L_2, \ldots, L_m$. When we apply probabilistic bounds, these inequalities may not be guaranteed. We will denote $E_j$ to be the event that $L_j$ is correct. Even though there is the chance of failure, we will operate assuming that all inequalities were correct. If they are indeed all correct for some $x$, then we can conclude that $f(x) \geq l^{(K)}$ for this particular $x$ as expected. Then it suffices to find a lower bound on the probability of the intersection of the $E_j$'s. We have

$$\Pr_{x \sim D}\left[f(x) \geq l^{(K)}\right] \geq \Pr_{x \sim D}\left[\bigcap_j E_j\right] = 1 - \Pr_{x \sim D}\left[\bigcup_j \overline{E_j}\right] \geq 1 - \sum_j \Pr_{x \sim D}\left[\overline{E_j}\right] = 1 - \sum_j q_j \tag{5}$$

as desired. $\qquad\qquad\qquad\qquad\qquad\qquad\qquad\qquad\qquad\qquad\qquad\qquad\qquad\qquad\square$

Now we will take advantage of this theorem. Assign every scalar inequality $i$ (either $A^L x + b^L \geq l$ or $A^U x + b^U \leq u$) a failure probability $q_i$ which sum to $Q$ altogether. We can invert the probabilistic inequalities to find scalars $l$ and $u$ such that these hold with given failure probability $q_i$. In particular, say that functions $\gamma_L(a, b)$ and $\gamma_U(a, b)$ are such that

$$\gamma_L(a, b) \leq \Pr_{x \sim D}[ax + b > 0] \leq \gamma_U(a, b)$$

for any $a, b$. We want to choose $l$ such that $\Pr[A^L x + b^L < l] \leq q_i$. Thus, it suffices to solve for $l$ such that $\gamma_L(A^L, b^L - l) = 1 - q_i$.

---

**Algorithm 1 I-PROVEN** for $B_\infty(x, \epsilon)$

---

1: **procedure** PRINEQ($x$, $\epsilon$, $A_L$, $b_L$, $A_U$, $b_U$, $q$)
2:  $\quad l_{\text{strict}} \to A_L x + b_L - \epsilon ||A_L||_1$
3:  $\quad u_{\text{strict}} \to A_U x + b_U + \epsilon ||A_U||_1$
4:  $\quad l_{\text{prob}} \to A_L x + b_L - \epsilon \sqrt{2 \ln \frac{1}{q}} ||A_L||_2$
5:  $\quad u_{\text{prob}} \to A_U x + b_U + \epsilon \sqrt{2 \ln \frac{1}{q}} ||A_U||_2$
6:  $\quad$ **return** $\max(l_{\text{strict}}, l_{\text{prob}}), \min(u_{\text{strict}}, u_{\text{prob}})$
7: **end procedure**
8: **procedure** IPROVEN($x$, $\epsilon$, $f$, $Q$)
9:  $\quad q \to Q/(2 \times \text{number of neurons})$
10: $\quad l^{(1)}, u^{(1)} \to$ PRINEQ($x, \epsilon, W_1, b_1, W_1, b_1, q$)
11: $\quad$ **for** i in [2, K] **do**
12: $\quad\quad A_L^{(i)}, b_L^{(i)}, A_U^{(i)}, b_U^{(i)} \to$ GETLINEARBOUNDS($l^{(:i)}, u^{(:i)}, A_L^{(:i)}, b_L^{(:i)}, A_U^{(:i)}, b_U^{(:i)}$)
13: $\quad\quad l^{(i)}, u^{(i)} \to$ PRINEQ($x, \epsilon, A_L^{(i)}, b_L^{(i)}, A_U^{(i)}, b_U^{(i)}, q$)
14: $\quad$ **end for**
15: $\quad$ **return** $l^{(n)}$
16: **end procedure**

---

For the uniform distribution $B_\infty(x, \epsilon)$, such functions $\gamma_L$ and $\gamma_U$ can be derived from Hoeffding's Inequality as seen in Corollary 3.2 of (Weng et al., 2019). Inverting them, we obtain

$$l = A_L x + b_L - \epsilon \sqrt{2 \ln \frac{1}{q_i}} ||A_L||_2. \tag{6}$$

Performing the forward and backward bound propagation methods ultimately yields a final scalar lower bound $l^{(n)}$. By theorem 1, we have that

$$\Pr_{x \sim D}[f(x) \geq l^{(n)}] \geq 1 - \sum_i q_i = 1 - Q.$$

We choose a simple scheme in assigning the failure probabilities. We select some subset of layers $S$ and assign all inequalities pertaining to these layers equal $q_i$'s. All other $q_i$'s are set to 0. So

$$q_i = \begin{cases} \frac{Q}{2 \times \text{ number of neurons within layers of S}} & \text{if } i \in S \\ 0 & \text{if } i \notin S \end{cases} \tag{7}$$

$q_i = 0$ indicates using the strict bounds and is only possible when $D$ has bounded support, as with $B_\infty(x, \epsilon)$. In this case, the Hölder inequality for the $l_\infty$ norm is used. Note that the original PROVEN algorithm is equivalent to only selecting the last layer.

We found that more complex assignment strategies such as optimizing over $q$ with sum equal to 1 did not lend themselves to any significant improvements compared to this simple method, especially given their additional run-time cost. However, there are a few factors to keep in mind when choosing the subset of layers $S$. In the case of uniform bounded noise, if the non-zero $q_i$ are small, it is possible for the strict inequalities, which find

$$l_{\text{strict}} = A_L x + b_L - \epsilon ||A_L||_1, \quad u_{\text{strict}} = A_U x + b_U + \epsilon ||A_U||_1 \tag{8}$$

to give tighter intervals than the probabilistic inequalities.

The different weight norms also create some discrepancy in the effectiveness of the probabilistic bounds. In particular, although the matrices $A$ in the above equations are not directly the network weights, we found that **I-PROVEN** performed worse on models with sparse weights. To take these into account, we return the tightest intervals using either the strict or probabilistic bounds, although we do not update the $q$'s to incorporate our choice.

## 3.2 OTHER DISTRIBUTIONS AND CERTIFIERS

**I-PROVEN can support distributions with infinite support.** The only requirement which **I-PROVEN** has on the distribution $D$ is that we must have lower and upper bounds on $\Pr_{x \sim D}[ax + b > 0]$ for $a, b \in \mathbb{R}$. This may include distributions with infinite support. For example, for additive iid Gaussian noise with standard deviation $\epsilon$, we can obtain

$$l = A_L x + b_L - \epsilon \operatorname{erf}^{-1}(1 - 2q_i) \|A_L\|_2 \tag{9}$$

from basic facts about Gaussian distributions. Similar formulas can be found for Gaussian mixtures when such a distribution is known and relevant. Note that $q_i$ must be non-zero for all inequalities when dealing with distributions with infinite support, as strict alternatives no longer exist.

**I-PROVEN can be used with any linear-relaxation-based certifier with no additional computational cost.** As **I-PROVEN** only requires changing the evaluation of the scalar bounds, it has *no* additional time complexity cost to whatever method it is being used with. It can be incorporated in any such linear-relaxation method. For a network with $K$ layers and $n$ neurons at each layer, the entire method when used with CROWN is $O(n^3 K^2)$ (Zhang et al., 2018), an additional factor of $K$ compared to a pass of the network.

## 3.3 PROBABILISTIC VERIFICATION BASED TRAINING

Our training method is simply a substitution of PROVEN into CROWN-IBP (Zhang et al., 2019). It requires two forward passes. The first computes strict bounds on each pre-activation neuron using IBP. The second performs linear bound propagations to compute linear bounds on the output. Then probabilistic bounds are applied to obtain the final scalar lower and upper bounds which are used in the loss function as described in CROWN-IBP's original paper. Note that this effectively means that only the original PROVEN is applied in this training method.

# 4 EXPERIMENTS

## 4.1 IMPLEMENTATION DETAILS

We performed experiments on the MNIST and CIFAR-10 datasets (LeCun et al., 2010; Krizhevsky, 2009). For our verification experiments, we used pretrained models provided in (Weng et al., 2018) and (Zhang et al., 2018) and the images from each dataset were scaled so that their values were in $[-0.5, 0.5]$. For our training experiments, we used the architectures described in (Gowal et al., 2019) and the images in the dataset were scaled to be in $[0, 1]$. Our MNIST models were trained for 100 epochs with a batch size of 100 and with a warm-up period for the first 5 epochs and a ramp-up period for the next 45. We followed the $\beta$ and $\kappa$ schedule used in (Zhang et al., 2019) for the PROVEN-IBP loss and we increase our $\epsilon$ from 0 to $4 \times 10^{-1}$ during the ramp-up period. We evaluate the validation set using $\epsilon = 3 \times 10^{-1}$. All code was written in Python with the use of the PyTorch library (Paszke et al., 2019) and training was conducted on a NVIDIA Tesla V100 GPU.

## 4.2 VERIFICATION RESULTS

We compare the original PROVEN bounds with our improved PROVEN on various MNIST and CIFAR-10 classifier in table 1. The classifier are $k$-layer MLPs with $n$ neurons in each layer, denoted as $k \times [n]$. In **I-PROVEN**, $q_i$ are all non-zero and equal across all layers. It achieves much better performance than the original PROVEN (Weng et al., 2019) with a 600% increase for 99.99% probabilistic robustness. Note that $Q = 0$ is simply the adversarial robustness certificate.

**I-PROVEN**'s improvement over PROVEN can be observed from intermediate layers' bounds. In table 2, the average interval gap of the neurons in a layer is tabulated. The model in particular is a 4-layer MNIST MLP (layer 0 indicates the input, which is why the interval length is simply $2\epsilon$). PROVEN and **I-PROVEN** are each evaluated on 10 images for various

Table 1: **I-PROVEN** versus original PROVEN for probabilistic robustness certificate on MNIST and CIFAR-10 models. **I-PROVEN** achieve significant improvements ($1.6\times$-$8.5\times$) on the tightness of probabilistic robustness certificate without additional computation cost.

| | $Q$ | 0 | 0.0001 | 0.01 | 0.05 | 0.25 | 0.5 |
|---|---|---|---|---|---|---|---|
| MNIST $2 \times [1024]$ | PROVEN | 0.0247 | 0.0325 | 0.0333 | 0.0337 | 0.0342 | 0.0345 |
| | **I-PROVEN** | 0.0247 | 0.0864 | 0.1013 | 0.1087 | 0.1181 | 0.1229 |
| | % improvement | 0 | **166** | **204** | **223** | **245** | **256** |
| MNIST $3 \times [1024]$ | PROVEN | 0.0194 | 0.0223 | 0.0226 | 0.0227 | 0.0228 | 0.0229 |
| | **I-PROVEN** | 0.0194 | 0.0640 | 0.0745 | 0.0797 | 0.0860 | 0.0893 |
| | % improvement | 0 | **187** | **230** | **251** | **277** | **289** |
| MNIST $4 \times [1024]$ | PROVEN | 0.0079 | 0.0083 | 0.0084 | 0.0084 | 0.0084 | 0.0084 |
| | **I-PROVEN** | 0.0079 | 0.0299 | 0.0347 | 0.0370 | 0.0398 | 0.0413 |
| | % improvement | 0 | **259** | **315** | **342** | **374** | **391** |
| CIFAR-10 $5 \times [2048]$ | PROVEN | 0.0017 | 0.0018 | 0.0018 | 0.0018 | 0.0018 | 0.0018 |
| | **I-PROVEN** | 0.0017 | 0.0123 | 0.0141 | 0.0150 | 0.0161 | 0.0166 |
| | % improvement | 0 | **574** | **673** | **720** | **776** | **804** |
| CIFAR-10 $7 \times [1024]$ | PROVEN | 0.0018 | 0.0018 | 0.0018 | 0.0018 | 0.0018 | 0.0018 |
| | **I-PROVEN** | 0.0018 | 0.0128 | 0.0147 | 0.0157 | 0.0168 | 0.0174 |
| | % improvement | 0 | **606** | **716** | **764** | **826** | **857** |

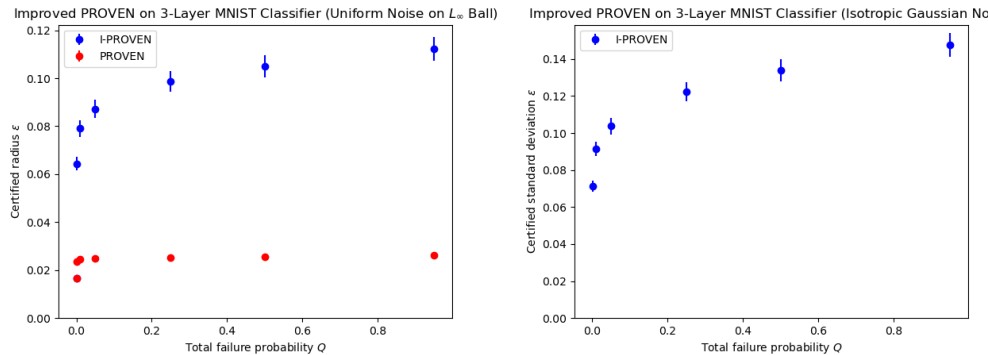

Figure 1: **I-PROVEN** verification results for a small 3-layer MNIST model considering the uniform distribution $B_\infty(x, \epsilon)$ and isotropic Gaussian distribution $N(x, \epsilon)$, respectively. Error bars are based on $3\times$ stdev.

$\epsilon, Q$. The scalar intervals $l, u$ which give bounds on each neuron in both methods are averaged within each layer. The smaller the interval size (that is, the tighter the interval), the better. **I-PROVEN**'s intervals are noticeably tighter from the first layer onwards as it invests a portion of the total failure probability $Q$ to tighten these earlier inequalities. This does mean PROVEN's final inequalities are sharper than **I-PROVEN**'s, which we can notice by observing the ratio of interval widths between Layer 3 and Layer 4. For $\epsilon = 0.001, Q = 0.1$ for example, PROVEN goes from 0.045 to 0.564, a $12.5\times$ increase, while **I-PROVEN** goes from 0.09 to 0.203, a $22.5\times$ increase. However, the improvements **I-PROVEN** makes in the earlier layers means it still ends up with tighter final bounds than PROVEN.

Note that we can our method is also able to handle additive Gaussian noise with only small changes to our probabilistic inequalities. Notably, we do not need to truncate the Gaussian distribution as (Weng et al., 2019) did. We plot our results for both additive uniform noise and additive Gaussian noise in fig. 1.

Table 2: Average interval lengths per layer

|  |  | Layer 0 | Layer 1 | Layer 2 | Layer 3 | Layer 4 |
|---|---|---|---|---|---|---|
| $\epsilon = 0.001, Q = 0.01$ | PROVEN | 0.002 | 0.049 | 0.043 | 0.045 | 0.564 |
|  | I-PROVEN | 0.002 | 0.011 | 0.009 | 0.009 | 0.203 |
| $\epsilon = 0.005, Q = 0.01$ | PROVEN | 0.010 | 0.247 | 0.324 | 0.522 | 7.478 |
|  | I-PROVEN | 0.010 | 0.054 | 0.052 | 0.055 | 1.146 |
| $\epsilon = 0.001, Q = 0.0001$ | PROVEN | 0.002 | 0.049 | 0.043 | 0.045 | 0.588 |
|  | I-PROVEN | 0.002 | 0.013 | 0.011 | 0.011 | 0.237 |

### 4.3 COMPARISON TO OTHER METHODS

In the earlier section, we compared I-PROVEN to PROVEN in terms of their robustness certificates. Now, we will show comparisons to IBP and a simple Monte Carlo approach based on (Anderson & Sojoudi, 2020). IBP does not generally perform well for arbitrary networks, so we use IBP-trained models in our experiment. In particular, we trained three CIFAR-10 CNN models A, B, and C. Model A was trained with standard loss, Model B was trained with an IBP loss term with $\epsilon$ ramping up to 2.2/255, and Model C was trained with an IBP loss term with $\epsilon$ ramping up to 8.8/255. We use two different $\epsilon$, 2/255 and 8/255, for each model and certifier. $Q$ is fixed at 1%. We apply these certifiers on 1000 images from the validation dataset.

For the Monte Carlo approach, we take $T \ln 1/u$ samples from the uniform $B_\infty(x, \epsilon)$ distribution where $u = 0.0001$ and $T = 100$, rounding to 921 samples per image $x$. Then if every sample is correctly classified, we conclude that at least $1 - 1/T = 99\%$ of the distribution is correctly classified with false positive rate less than $u$. This Monte Carlo approach has a higher false negative rate than other sampling approaches, but we chose this one beecause it requires the lowest number of samples as far as we are aware.

Table 3: IBP, PROVEN, **I-PROVEN**, and the Monte Carlo method on a CIFAR-10 classifier. The entries indicate the portion of images which each certifier is unable to certify as robust (lower is better). Standard (error) denotes the portion of images which is classified incorrectly without any perturbations.

|  |  | IBP | PROVEN | I-PROVEN | Monte Carlo | Standard |
|---|---|---|---|---|---|---|
| $\epsilon = 2/255$ | Model A | 1.000 | 0.996 | 0.808 | 0.429 | 0.409 |
|  | Model B | 0.715 | 0.643 | 0.598 | 0.511 | 0.483 |
|  | Model C | 0.621 | 0.604 | 0.613 | 0.576 | 0.552 |
| $\epsilon = 8/255$ | Model A | 1.000 | 1.000 | 1.000 | 0.491 | 0.409 |
|  | Model B | 0.993 | 0.978 | 0.952 | 0.588 | 0.483 |
|  | Model C | 0.806 | 0.797 | 0.801 | 0.634 | 0.552 |

As the results in table 3 show, I-PROVEN performs better than IBP across all models, but this gap diminishes greatly in the two IBP-trained models, particularly for Model C. Similarly, we see the gap between PROVEN and I-PROVEN close and PROVEN even outperforms I-PROVEN in Model C. We see a similar phenomena in the next section, section 4.4, and provide an explanation.

Unsurprisingly, the Monte Carlo approach obtains better results than any of the relaxation-based approaches. Furthermore, in terms of timing, PROVEN/I-PROVEN took 128-131s for all 1000 images per model, while the Monte Carlo approach consistently took around 4s. Evaluating the standard error and IBP error took under a second. However, the exact details on timing do depend somewhat on the situation. In this experiment, **I-PROVEN** could not be batched at all as the memory used was too expensive, while Monte Carlo methods can easily be batched.

### 4.4 A CASE STUDY ON TRAINING WITH I-PROVEN

We examine **I-PROVEN**'s verification compared with IBP's on models trained with IBP and PROVEN-IBP, respectively, on the small CNN from (Gowal et al., 2019) and the verification algorithms are considering $B_\infty(x, 0.3)$. Note that IBP is considering the model's adversarial robustness while **I-PROVEN** is considering the probabilistic robustness for $Q = 1 \times 10^{-2}$ (equivalently, for 99% of the ball). We found that **I-PROVEN** does not obtain significantly better results than IBP in either case, mirroring our results on the IBP-trained models in table 3. We conjecture that this is due to the weight sparsity induced by IBP's involvement in the training for both pure IBP training and PROVEN-IBP, and that this sparsity is also present in the linear bounds for **I-PROVEN**. Weight sparsity is beneficial for adversarial robustness, particularly for the $l_\infty$ norm (Xiao et al., 2018). However, as far as we are aware, there is no reason to expect weight sparsity to help a model's probabilistic robustness and as noted in section 3.1, **I-PROVEN**'s probabilistic inequalities prefer more evenly distributed matrices.

This weight sparsity also explains why PROVEN outperforms **I-PROVEN** in Model C of table 3. When the model weights are sparse, the strict inequality is tighter than the probabilistic inequalities for very small $q_i$ and so **I-PROVEN** does not perform as well as usual by distributing $Q$ evenly among the inequalities.

To test our hypothesis, we compared the last layer's linear lower bounds $A_L$ between a CNN trained in a standard manner, and a CNN trained with an IBP loss as in (Gowal et al., 2019). We show this for an image from the validation set in fig. 2. These $A_L$'s came from **I-PROVEN** applied with $\epsilon = 0.3$, $Q = 1 \times 10^{-2}$, and the failure probabilities in the last two layers non-zero and equal. Each $28 \times 28$ grid corresponds to a logit in the output. The grid for 3 is all 0 as this is specifically considering the margin function between each class with the true class, 3. The absolute values of the matrix are scaled to fit in $[0, 1]$. Evidently, the values from the IBP model are far more sparse. Further examples are included in the appendix.

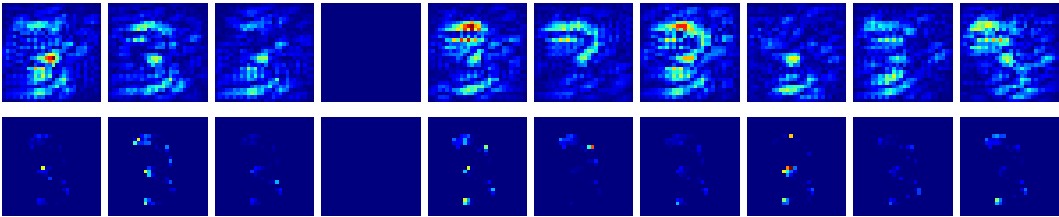

Figure 2: $A_L$ from a CNN trained in a standard manner (top row), and a CNN trained with IBP respectively (bottom roow).

## 5 CONCLUSION

In this paper, we present **I-PROVEN**, an algorithm that can efficiently verify the probabilistic robustness of a neural network. We show strong improvements compared to the prior method used against this problem: we remove the assumptions of bounded support and significantly improve the tightness of robustness certificate without any additional cost. Furthermore, we present a training framework for probabilistic robustness and demonstrate its shortcomings. By taking a closer look at these results, we make steps towards understanding the relation between adversarial certified defense methods and our own.

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
