# OpenReview forum: "Efficient Certification for Probabilistic Robustness"
_ICLR.cc/2022/Conference — ICLR 2022 Submitted_

### Official Review · Reviewer_JweR · 2021-10-30

**Correctness:** 3
**Technical Novelty And Significance:** 1
**Empirical Novelty And Significance:** 2
**Recommendation:** 5
**Confidence:** 4

**Main Review:**

The main result, Theorem 1, seems rather trivial. While it is good to highlight that one can consider a union over all layers, the contribution may not be significant enough for ICLR.

***updates after author response***

I am still feeling that, while the observation of union bound is cute, the overall technical contribution is below the bar (the union bound is the only new analysis). However I do encourage authors to further explore the power of the bound, likely among other things, to get a stronger algorithm.

**Summary Of The Paper:**

see below

**Summary Of The Review:**

see above.

---

> ### Author Response · Authors · 2021-11-23
> **Thanks for your feedback! Please also check the general comments.**
>
> Thank you for your helpful feedback! We’ve included some further context and additional experiments in the general comment.

---

### Official Review · Reviewer_5NRz · 2021-11-02

**Correctness:** 4
**Technical Novelty And Significance:** 2
**Empirical Novelty And Significance:** 2
**Recommendation:** 5
**Confidence:** 4

**Main Review:**

Overall, the paper seems to be an incremental improvement over PROVEN algorithm proposed by Weng et al.
While I like the general approach of combining several linear relaxations with union bound, and thus improving the
results, I also have some concerns, which I list below.

1) I am not sure that the threat model considered in this work is that relevant.
I think authors should explain what would be some setting in which adversary is performing these probabilistic local perturbations.
Right now, it seems to me that the setting is quite artificial, and there does not seem to be that much interest in the literature
in solving this problem, except the work of Weng et al. Could authors motivate their problem setting better?

2) In the background, authors mention some methods based on Chernoff bound, and argue that these methods need high number of samples to achieve a high degree of accuracy. Why are these methods, e.g. the one proposed by Baluta et al., not compared to in the experimental evaluation? Without this, I think it is not actually clear whether these methods do not scale.

3) Could authors discuss limitations of their approach? Right now it seems that, as all of the convex relaxation based approaches, this approach does not scale to large networks, while e.g. Chernoff bound based approach is essentially independent of the architecture.

4) Could authors discuss relationship between the proposed method and randomized smoothing [1, 2]? Could randomized smoothing be applied to the threat model of probabilistic robustness that authors consider in this work? Fundamental limitation of approaches based on convex relaxations is that they do not scale to large networks, and randomized smoothing has worked well for achieving provable (deterministic) local adversarial robustness even on large networks and datasets, so it would be interesting to know whether it can work in this setting as well. Especially given the fact that guarantees are already probabilistic, randomized smoothing might be better tool to solve this problem than convex relaxation based approaches.

Typos:

- "Note that we can our method"

[1] Lecuyer, Mathias, et al. "Certified robustness to adversarial examples with differential privacy." 2019 IEEE Symposium on Security and Privacy (SP). IEEE, 2019.

[2] Cohen, Jeremy, Elan Rosenfeld, and Zico Kolter. "Certified adversarial robustness via randomized smoothing." International Conference on Machine Learning. PMLR, 2019.


**Summary Of The Paper:**

This work considers problem of local robustness where each input is perturbed according to some probability distribution (e.g. uniform distribution over the L-infinity ball). Proposed approach is based on extending an approach proposed by prior work which uses linear bounds to compute bounds on the output of the network. The key contribution is computing the probabilistic bounds for inner layers, and not only the final one as was done in prior work. Authors show that their approach improves over prior work on MNIST and CIFAR-10 datasets.


**Summary Of The Review:**

I find this paper marginally below the acceptance bar because the problem setting is not that well motivated, and the approach itself is incremental over the prior work PROVEN.

---

> ### Author Response · Authors · 2021-11-23
> **We give clarifications regarding the context of our work and provide additional experiments comparing our method to Monte Carlo methods.**
>
> Thank you for your helpful feedback!
>
> ### Motivation
>
> The specific hyperrectangle threat model used in Section 3 is used to illustrate the method and as a baseline for comparisons. It is not meant to model common threats, but as noted in Section 3.2, our method can be used for more applicable threat models such as additive white Gaussian noise. It has been pointed out in several papers that image classifier models perform worse than humans for this simple threat.
>
> Some motivation is also derived from this work’s connection to randomized smoothing, as you observed in your fourth point. Randomized smoothing uses a base classifier with some probabilistic robustness guarantees and from it builds an ensemble with adversarial/strict robustness guarantees. Then the problem of training an adversarially robust classifier could possibly be solved by training a probabilistically robust classifier. Ideally, we could use our method in a training loss but as described in Section 4.4, there are obstacles.
>
> ### Sampling methods
>
> This is a good point. We have included such experiments in the general comments. Such methods outperform I-PROVEN in terms of the ability to certify, but can be less efficient, especially as the number of samples is highly dependent on Q.
>
> ### Limitations
>
> You are correct, our method shares the same limitations as other convex-relaxation-based methods. In particular, the performance falls off greatly for sufficiently deep networks as the errors from performing the convex relaxations compound. It should be noted that this fall-off is dependent on how tight the convex relaxation bounds are.
>
> ### Randomized smoothing
>
> As mentioned above, our method could be used to give certifications on the probabilistic robustness of the original classifier (i.e. the *base* classifier in the randomized smoothing paper), while randomized smoothing is changing the prediction rule so that the certification guarantees are on the *smoothed* classifier, which has a totally different purpose. Moreover, our problem setting considers the threat model that has natural/random noises and I-Proven could provide probability guarantees on the robustness certificate of the *original* classifier. On the other hand, randomized smoothing consider the threat model that has adversarial/worst-case perturbation, and the robustness certificate is on the *smoothed* classifier.
>
> ### Summary
>
> We have included further comparisons against simple sampling methods. We have also given clarifications based on your suggestions.

---

> > ### Comment · Reviewer_5NRz · 2021-11-29
> > **Response**
> >
> > Thank you for your response. After reading other reviews and the response, I decided to keep my score.
> >
> > I think that providing comparison with sampling methods is definitely good addition to the paper as it provides us much better picture of
> > the trade-offs that the method makes. I like that you highlight both positives and negatives of approaches based on sampling and convex relaxations, e.g. sampling can work for any architecture or distribution, but it requires more time. However, based on the results in your general response, I think it is hard to say whether I-PROVEN or sampling would be more useful in practice. There is a drastic difference in certification rate -- sometimes Sampling certifies 10x more, and one might argue that this increase in guaranteed safety is worth the runtime overhead. Overall, I would suggest authors to focus on the motivating their approach better by providing some more concrete use cases where using sampling is not possible due to the runtime overhead, and where we can better see the usefulness of I-PROVEN.

---

### Official Review · Reviewer_o2Ha · 2021-11-03

**Correctness:** 3
**Technical Novelty And Significance:** 3
**Empirical Novelty And Significance:** 2
**Recommendation:** 5
**Confidence:** 3

**Details Of Ethics Concerns:**

None for this paper.

**Main Review:**

I think the pros of this paper are its clear exposition, goals, and methodology. In particular, the problem of adversarial examples is a serious one and continuing to work on expanding adversarial robustness guarantees to larger models is a avenue of research that can have clear impact. Further the goals and method of the paper are clearly explained. Finally, the proposed algorithm does have some key empirical advantages over the PROVEN algorithm, often gaining significantly better certified radii.

The cons of this paper lie in questions of its practicality compared not just to the PROVEN algorithm but to sampling based methods. For both PROVEN and I-PROVEN I find the use of convex relaxation a questionable choice due to several limitations. Firstly, these relaxations are known to introduce non-trivial over-approximations in the output. This is, of course, acceptable when performing rigorous safety verification, but given that the guarantees desired in this work are only statistical/probabilistic performing such relaxations (especially on large models on CIFAR) might introduce unnecessary over-approximation which leads to vastly conservative estimates in safety radius. I would be greatly surprised if this method was able to produce better radius than statistical methods given that they do not introduce such over approximations. Thus, there is an unexplored trade-off in a few directions that I think limits the impact of this paper. In particular, my questions are (1) how much more costly is this method (in terms of computational complexity and time) than the standard PROVEN algorithm? Just above subsection 3.3 they note its complexity relative to CROWN and I suspect that this complexity comparison will hold compared to PROVEN (that it is only a linear factor slower); however, it would be good to get an idea of how much slower this is in practice for their large CIFAR networks. (2) How much tighter are the statistical bounds? I expect them to be a great deal tighter, but also much more expensive to compute. It would be interesting to see what the trade off is here. Clearly I-PROVEN should theoretically fall in between PROVEN and sampling methods in terms of its tightness and its computational time, yet the authors do not explore this.

Moreover, I think the extension to infinite support noise distributions is a rather weak extension. Clearly, PROVEN can also truncate a Gaussian and make the same erf function argument that is used here without any issue, so I do not see this as an extension unique to the proposed methodology. Of course, for statistical methods, infinite support is just fine, and so are poorly defined densities (i.e. distributions which can only be efficiently sampled from) which is something this method cannot support and this should be noted as a limitation.

Finally, I think it would be interesting to compare the training method proposed here to randomized smoothing of classifiers and the MACER algorithm for training of robust randomized smoothing classifiers.





**Summary Of The Paper:**

In this paper, the authors consider a notion of statistical/probabilistic robustness which does not require a model to be robust to all inputs in a specified set, only a certain, high-probability subset of these inputs. The authors rely on a bound propagation methodology to compute the the probability that a given input property is violated. In particular, the authors expand a known methodology (PROVEN) for computing probabilistic robustness and show that in many cases their methodology is better than that of PROVEN.

**Summary Of The Review:**

I think this is a good paper and is heading in the right direction, but needs a bit more work to realize its potential impact. In particular, I believe it needs a bit more discussion and evaluation in order to fully understand where this method falls in relation to previously proposed methods.

---

> ### Author Response · Authors · 2021-11-23
> **We clarify that the runtime of I-PROVEN is the same as the wrapper certifier (CROWN in our experiments) and provide additional experiments comparing our method to Monte Carlo methods.**
>
> Thank you for your helpful feedback!
>
> ### Limitations of convex-relaxation-based methods
>
> You are correct, our method shares the same limitations as other convex-relaxation-based methods. In particular, the performance falls off greatly for sufficiently deep networks as the errors from performing the convex relaxations compound. It should be noted that this fall-off is dependent on how tight the convex relaxation bounds are.
>
> ### Runtime
>
> To clarify, CROWN, and the implementation of PROVEN and I-PROVEN in their respective papers all have the same time complexity and runtime. If I-PROVEN were used with a different convex-relaxation-based method, then it would match its runtime. Please let us know if Section 3.2 is unclear on this.
>
> ### Sampling methods
>
> We have also included a comparison with sampling methods for the certified radius in the general comments.
>
> ### Summary
>
> We clarify the runtime of I-PROVEN and provide further comparisons between I-PROVEN and sampling-based methods.

---

### Official Review · Reviewer_sAmC · 2021-11-03

**Correctness:** 2
**Technical Novelty And Significance:** 2
**Empirical Novelty And Significance:** 2
**Recommendation:** 3
**Confidence:** 2

**Main Review:**

Strengths: An interesting problem, and improves over a previous method.

Weaknesses: Overall, I found the empirical results unconvincing because they compared only to a single existing method (which was essentially a worse version of the current method). A simple baseline would be to use a stronger verification method (e.g. https://arxiv.org/pdf/2010.11645.pdf) with Q=0, which might actually outperform the proposed method even allowing for larger Q. There are also methods for related problems that should be able to handle relatively small Q (perhaps smaller than those considered in the current paper), such as this paper: https://arxiv.org/abs/2008.10581. It is difficult to contextualize the results without comparisons or at least discussions of these methods.

**Summary Of The Paper:**

The paper considers the problem of probabilistic certification of robustness, that is, showing that the probability that a random point in some hyperrectangle changes the classification. It improves an existing method, PROVEN, to produce a new method I-PROVEN that it shows achieves better empirical results.

**Summary Of The Review:**

Overall, the experiments are too narrow to support the main claims, so I recommend against accepting the paper.

---

> ### Author Response · Authors · 2021-11-23
> **We provide comparisons to Monte Carlo methods and clarify that I-PROVEN can be used with stronger linear-based certifiers**
>
> Thank you for your helpful feedback! We’ve responded to your feedback below. We would also appreciate it if you explained why you gave the Correctness a 2.
>
> ### Comparisons to other methods
>
> Please see the general comment for a comparison with sampling methods. The takeaway is that sampling methods can certify more distributions, but require a longer runtime for small failure probabilities $Q$, even for works which are specifically for small $Q$. For stronger adversarial robustness certifiers, we reiterate that our method can be used in any linear propagation model including BetaCROWN [1] which is state of the art.
>
> [1] Wang, Shiqi, et al. "Beta-crown: Efficient bound propagation with per-neuron split constraints for neural network robustness verification." Advances in Neural Information Processing Systems 34 (2021).

---

> > ### Comment · Reviewer_sAmC · 2021-11-29
> > **No all verification methods are Monte Carlo**
> >
> > Thanks for your response. I am not convinced by the additional experiment, partly because there are many verification methods that are not Monte Carlo--for instance the first paper that I linked to above, which uses semidefinite programming instead and is deterministic.
> >
> > Regarding correctness, the 2 category refers to "Several of the paper’s claims are incorrect or not well-supported." Since the baselines were insufficient, I did not consider the empirical claims to be well-supported. Hope that helps to clarify!

---

### Decision · Program_Chairs · 2022-01-20

**Decision:**

Reject

**Comment:**

The paper proposes a method to improve PROVEN, which gives a certification for probabilistic robustness. However, reviewers think the paper is below the acceptance bar due to unclear motivation and insufficient experiments. In particular, a clear use case of probabilistic robustness certification is crucial for the paper.